# The Impact of Plant Essential Oils and Fine Mesh Row Covers on Flea Beetle (Chrysomelidae) Management in Brassicaceous Greens Production

**DOI:** 10.3390/insects11100714

**Published:** 2020-10-19

**Authors:** Robert Brockman, Ryan Kuesel, Kendall Archer, Kyla O’Hearn, Neil Wilson, Delia Scott, Mark Williams, Ricardo Bessin, David Gonthier

**Affiliations:** 1Department of Entomology, S-225 Ag. Sci. Center North, University of Kentucky, Lexington, KY 40546, USA; rwku222@g.uky.edu (R.K.); Kendall.Archer@uky.edu (K.A.); kboh223@g.uky.edu (K.O.); delia.scott@uky.edu (D.S.); Ric.Bessin@uky.edu (R.B.); 2Department of Horticulture, N-322 Ag. Sci. Center North, University of Kentucky, Lexington, KY 40546, USA; neil.wilson@uky.edu (N.W.); mark.williams@uky.edu (M.W.)

**Keywords:** organic agriculture, arugula, mizuna mustard, row cover, ProtekNet, Agribon, flea beetle, *Phyllotreta striolata*, *Phyllotreta bipustulata*, *Phyllotreta cruciferae*, *Chaetocnema concinna*

## Abstract

**Simple Summary:**

The popularity of brassicaceous leafy greens has grown in recent years due to health benefits and the local food movement. However, for many producers and especially organic producers, flea beetles represent a major challenge to production because they chew small holes in the leaves reducing quality and appearance. The goal of our project was to assess the efficacy of row covers and essential oils in controlling flea beetles as an alternative to organic and conventional insecticides. We grew Arugula and Mizuna mustard greens in replicated experimental plots in the spring and fall of 2019. We found that in most cases, plants within the Agribon and ProtekNet row cover treatments had the least amount of flea beetle damage and the highest yields, while plants within the essential oil treatments and organic insecticides did not differ from unsprayed, uncovered control plots. Conventional insecticides had an intermediate level of control against flea beetles, but did not perform as consistently as row cover treatments. We believe that row covers could provide a strong management method for all growers of brassicaceous leafy greens, especially where quality standards are high.

**Abstract:**

Brassicaceous leafy greens are an important crop for small growers but are difficult to produce due to damage by flea beetles. Flea beetles are problematic for growers as they chew many small holes through leaves rendering produce unmarketable. We tested the efficacy of several essential oils, the woven-mesh row cover ProtekNet, and the spunbonded row cover Agribon, compared to organic and conventional insecticides and no spray controls in the spring and fall of 2019. We found that the two row cover treatments (Agribon and ProtekNet) provided the best control of flea beetles and associated damage. Thyme oil was highly phytotoxic and killed the crop entirely and rosemary and neem essential oils caused mild phytotoxic burns. Organic insecticides rarely performed better than the no spray control. While conventional insecticides controlled most flea beetles, the crop was often still too highly damaged to sell. The results of our study suggest row covers offer producers an effective method of flea beetle control that reduces their dependence on insecticides for conventional and organic production.

## 1. Introduction

Brassicaceous leafy greens (*Brassicaceae*) are an economically important and micro-nutrient rich crop grown in many parts of the world. Brassicaceous leafy greens are termed a specialty crop [1], and within Kentucky, are grown on a small scale and sold directly to consumers. Farmers can grow brassicaceous greens crop in the fall and spring to elongate their offerings at farmer’s markets. The demand for local fresh produce has grown dramatically in the last two decades. Much of the interest in local foods has emerged as consumers believe that the food is fresher, of higher quality, and healthier [2]. Leafy greens are nutrient dense and rich in antioxidants [3] making them popular with health-conscious consumers. Eighty five percent of Community Supported Agriculture businesses believe that demand for local food is increasing [4] and farmer’s markets grew from 1755 in 1994 to 8687 in 2017 [5]. Many consumers are willing to pay large price premiums to buy produce that is local [6,7] and even more for produce that is both locally grown and organic [8]. The US organic food industry has grown from 9.6 billion dollars in 2003 to 47.9 billion dollars in 2018 [9]. Within Kentucky, leafy greens sold for an average of $3.15/pound between 2014 and 2018 [10].

Despite this demand, brassicaceous leafy greens are difficult to produce due to heavy pest pressure from flea beetles (Coleoptera: Chrysomelidae: Alticini) that can render the leaf tissue unmarketable. Adult beetles feed on the foliage which leads to direct damage to produce. The damage associated with flea beetles is very distinctive with many small holes spread throughout the leaves which are termed “shot holes”. Flea beetles owe their common name to their saltatorial hind legs which are used for jumping long distances and escaping from predators. Additionally, the small size of the flea beetles allows them to hide within leaves where they are protected from contact with foliar insecticides. Flea beetles spend a portion of their lives underground which can further complicate control as their eggs, larvae, and pupae are protected from insecticide sprays. Flea beetles start their lives as eggs laid on the surface of their host plant’s roots [11]. These eggs hatch and go through three larval instars before pupating and emerging as adults. This portion of the life cycle takes 44 to 55 days in the Canadian Great Plains for species within the problematic genus *Phyllotreta* [11,12]. In the northern portion of the Canadian Great Plains, most species only have one generation per year while in the southern portion of the Canadian Great Plains and the New England area, two generations are common [11]. Most of the damage to the crop occurs during the beetle’s adult life stage with only minimal damage caused by larval feeding on roots [11]. Currently, farmers raising brassicaceous leafy greens rely primarily on conventional insecticide treatments [13]. 

While conventional insecticides are the most common method of flea beetle control, research suggests that they are relatively ineffective at controlling flea beetles. A study comparing the efficacy of insecticides in Canada found that the effects of thiamethoxam and pyrethrin were not statistically different from the control. However, carbaryl, the most effective synthetic insecticide treatment, was better than the unsprayed control [14]. Furthermore, Walgenbach and Schoof [15], found that five classes of systemic insecticide did not reduce flea beetle damage compared to no spray controls. Only at high concentrations did application of cyantraniliprole result in lower flea beetle damage [15]. For organic growers, organic insecticides show some success at reducing flea beetle abundance and damage [16]. Andersen et al. [14], found that an organic spinosyn mix reduced flea beetle damage compared to the control. The poor effectiveness of many insecticides may be due to the development of resistance in some flea beetle populations. For example, Canadian populations of the crucifer flea beetle, *Phyllotreta cruciferae* [17] and European populations of cabbage stem flea beetle, *Psylliodes chrysocephala* L. [18] have developed resistance to insecticides. Furthermore, conventional, and organic insecticides have non-target effects on beneficial insects such as pollinators and natural enemies [19]. Thus, integrated pest management programs (IPM) often promote the use of alternative non-chemical practices in place of insecticides.

A number of alternative practices for controlling flea beetles have been evaluated, including: removing old crop debris during winter, planting trap crops, modifying planting date, plant-derived essential oils, and row covers [13]. These alternatives vary in success rates and feasibility of implementation. For example, flea beetles overwinter as winged adults which can move across fields easily [20], making removal of crop residues unlikely to be effective. Trap cropping systems for brassicaceous greens may be difficult to implement given that greens growers often have complex planting schedules and little extra time or space, making incorporation of trap crops difficult [21]. Modifying planting dates to avoid flea beetles is problematic as high temperatures prevent leafy greens production during mid-summer for much of the southern U.S. and other regions. While trap cropping and modifying planting dates may be difficult for brassicaceous greens growers to implement, many of these alternatives show promise for the future. 

Recently, the use of plant-derived essential oils that act as insect repellents has been promoted as a safer alternative to synthetic insecticides. As plant essential oils are natural products and are often extracted from culinary herbs, they are viewed as safe to eat [22] and are believed to have minimal negative effects on beneficial insects [22,23]. Laboratory experiments have revealed that essential oils such as peppermint (*Mentha* × *piperita*), thyme (*Thymus vulgaris*), rosemary (*Salvia rosmarinus*), cloves (*Syzygium aromaticum*), Norwegian angelica (*Angelica archangelica*), and basil (*Ocimum basilicum*) have repellent effects on spotted wing drosophila [24], aphids [23,25,26,27], and Colorado potato beetle [28,29]. Essential oils are volatile and have poor water solubility which make them difficult to use in agricultural settings [30]. Some aromatic and culinary herbs have essential oils which are phytotoxic and give them a competitive edge over weeds [31]. Studies involving coriander (*Coriandrum sativum*), tobacco (*Nicotiana tobacum*), and fenugreek (*Trigonella foenum-graecum*) oils found no phytotoxic effects to apple, rose, and oleander plants [26,27]. However, relatively little research has examined the phytotoxicity of many essential oils. Regardless, pest control companies have started to market essential oil mixes. Commercial essential oil mixes include Trifecta Crop Control (Trifecta LLC, South Williamsport PA, USA), Essentria (Zoecon, Schaumburg IL, USA), and Ant Out (JH Biotech Inc., Ventura CA, USA). Few studies on essential oils have tested the longevity of repellent effects under field cropping conditions. Furthermore, few field studies on flea beetles have been published in the scientific literature.

Perhaps the most effective alternative flea beetle control tactic is the use of row covers [14,32]. Row covers are light fabrics or netting that cover the crop to form a physical barrier to a pest insect. The use of row covers in vegetable production was first introduced for season extension through the use of spun-bonded or perforated polyethylene blankets that raise the temperature 2–7 °C [33,34]. However, these same covers can be applied to limit insects’ access to vegetable crops. Furthermore, insect exclusion row covers have successfully limited pests in broccoli [35], grape [36], apple [37], squash [35,38], muskmelon [38], blackberry [39], blueberry [40,41], and raspberry [42,43]. ProtekNet, made of knitted polyamide, provides a maximum increase in temperature of only 0.5 °C and a maximum decrease of 5% humidity [37]. ProtekNet also protects crops from diseases and extreme weather events such as hail [37]. To our knowledge only one study has looked at the efficacy of row covers for control of flea beetles in brassicaceous greens. Andersen et al., [14] found that the row cover CovertanPro 30 reduced flea beetle abundance and damage more than any conventional and organic insecticide treatment. While Andersen et al., [14] found strong results in Massachusetts, few growers have adopted row covers in the southeastern United States. Questions remain if row covers can be successful in the heat of the south.

Given the challenge of controlling flea beetles in leafy brassicaceous greens, it is imperative to develop best management systems by comparing and combining alternative practices. In this study, we compare the efficacy of the woven-mesh row cover ProtekNet, the poly spunbonded row cover Agribon AG-30 (hereafter known as Agribon), several plant-derived essential oils, and organic and conventional insecticide treatments in both arugula and mizuna mustard greens. We hypothesized that row covers would increase quality by reducing flea beetle damage to leafy greens. We expected the plots treated with row covers and conventional insecticides to have higher yields than all other treatments. Furthermore, we hypothesized that combining plant essential oil sprays and row covers would maximize flea beetle control. 

## 2. Materials and Methods

Site: Field studies were conducted during the spring and fall of 2019 at the University of Kentucky’s Horticulture Research Farm, located in Lexington, Kentucky (37°58′25.92″ N, 84°32′5.85″ W). This 100-acre farm is within plant hardiness zone six. The farm is split into organic and conventional zones and includes a diverse arrangement of crops. Our field was located next to fields growing clover, tomatoes, and radishes in the conventional section of the farm. To study the impact of row covers, essential oils, and organic and conventional insecticides, we performed two field trials grown on a 0.275-acre field site of arugula (Astro) and mustard greens (Mizuna) in the spring (11 April–20 May) and fall (16 August–20 September) of 2019. In Kentucky, high temperatures in the mid-summer limit the success of mustard greens. For this reason, we focused on spring and fall plantings.

A randomized block design with a split plot was used for both field trials in 2019. Each plot consisted of three raised beds with two rows being planted in each raised bed. The rows within each bed were 18 inches apart and seed was spaced 0.6 inches apart within row. Within each bed, one row of arugula was grown and one row of mizuna mustard was grown. The outer two raised beds were established and treated identically but were used as guard rows with no data collection. All data were taken from the center four feet of the raised bed to provide a three-foot buffer from the edge of the plot in all directions. 

Spring trial field design: there were seven treatments for each field trial that varied between the spring and fall trials (Table 1). These treatments were replicated four times among four blocks for the spring and fall trials. The treatments were as follows: (1) control treatment. (2) The organic insecticide treatment. (3) The conventional insecticide treatment. (4) The Agribon treatment fully covered the plot from the time of cotyledon emergence until harvest. (5) The ProtekNet treatment fully covered the plot from cotyledon emergence until harvest. (6) The ProtekNet + rosemary (Salvia Rosmarinus) oil treatment. (7) The ProtekNet + thyme (*Thymus vulgaris*) oil treatment.

Field preparation and treatment implementation: in 19–22 March 2019, the field was disked, and compost was applied at a rate of 10 tons per acre. On 24 March, the field was spaded to incorporate the compost, beds were formed, and two lines of drip tape were buried per bed (Aqua-Traxx 6″, Toro Garden Company, Bloomington, MN, USA). The field was then shallowly cultivated on 4 and 11 April to form a stale seed bed. Arugula and mizuna mustard were planted on 11 April at 0.6 inch spacing. All seed was sourced from Johnny’s Selected Seed (Winslow, ME, USA). Mizuna was planted in the northern row of each bed while arugula was planted in the southern row of each bed. Nature Safe 10-0-8 (Darling Ingredients Inc., Irving, TX, USA) was applied at a rate of 50 lbs.(pounds) N per acre. All plots were planted on bare ground and we manually weeded as needed to suppress weeds.

Row covers were implemented shortly after germination on 23 April. Essential oils were sprayed twice a week at a rate of 500 mL per ten feet of row. We selected this rate and frequency to test the maximum realistic rate for our field study. We anticipated that this high rate would not be used for market farming systems, and we anticipated lowering the rate for future studies. Flea beetles were first observed on 29 April which initiated the weekly insecticide spray schedule. Plots were sprayed a total of three times with insecticide treatments. Organic insecticides were rotated between Entrust SC (Spinosad, Corteva Agriscience [Dow AgroScience], Indianapolis, IN, USA) and Pyganic 5.0 (Pyrethrin, Valent U.S.A. Corporation, MGK, Minneapolis, MN, USA), while the conventional insecticides were rotated between Mustang Maxx (Pyrethroid, Zeta-cypermethrin, FMC Corporation, Philadelphia, PA, USA) and Scorpion 35SL (Dinotefuran, Gowan Company, Yuma, AZ, USA). Insecticides were applied at the industry recommended concentrations. These rates were 5 oz/acre for Scorpion 35SL, 6 oz/acre for Entrust, 2.75 oz/acre for Mustang Maxx, and 8 oz/acre for Pyganic 5.0. At the conclusion of the spring trial, we seeded buckwheat as a cover crop for four weeks. This cover was flail mowed and incorporated into the soil on 8 August. 

Fall trial field design: the experimental design of the fall trial differed from the spring trial only in terms of the essential oil treatments. Here, seven treatments were replicated four times among four blocks (Table 1). The treatments were as follows: (1) control treatment; with no spray and no row covers. (2) The organic insecticide treatment. (3) The conventional insecticide treatment. (4) The Agribon treatment. (5) The ProtekNet treatment. (6) The rosemary oil treatment. (7) The neem (*Azadirachta indica*) oil treatment. 

*Field preparation and treatment implementation:* the same field used for the spring trial was used for the fall trial. Field preparation for the field trial was the same as the spring trial. The field was cultivated on 15 August and the beds were formed with the stale seed bedding attachment. Two lines of drip tape were buried in each bed on 16 August and beds were reformed. Arugula and mizuna mustard were planted on August 16th at a 0.6 inch spacing. Mizuna was planted in the northern row of each bed while arugula was planted in the southern row of each bed. 

Seedlings started to emerge on 20 August and row covers were implemented on that day. Essential oils were sprayed twice a week. For the first week only, both essential oils were sprayed at a rate of 60 mL per 10 ft (feet) or 180 mL per plot. For the rest of the trial, essential oils were sprayed at a rate of 125 mL per 10 ft or 375 mL per plot. The choice to lower application rates in the second trial was made to match the difference in surface area requiring spray coverage. In the fall trial, spray coverage was only needed to cover the brassicaceous greens’ foliage whereas in the spring trial, we sprayed the entire netted canopy. Insecticides were sprayed weekly starting on 27 August. Organic and conventional insecticide treatment plots were sprayed a total of four times following the same rotations outlined in the spring trial.

Data collection:

*Insect pest monitoring data:* all insect pests were sampled during the season using yellow sticky traps (Arbico Organics, 5″ × 7″ Yellow Sticky Traps, Tucson, AZ, USA). Yellow sticky traps were placed in the field twice during each growing season. During each of the two time periods, yellow sticky traps were left in the field for one week before being collected. For each time period, a standard 5″ × 7″ yellow sticky trap was cut in half and these two halves were placed within the plot. As an additional measurement to determine the relative abundance of flea beetle species, we made collections by vacuum sampling with an inverted leaf blower before harvest in the spring and fall (STIHL 5H 56C, STIHL, Inc., Waiblingen, Germany). We modified protocols from Swezey et al. [44]. Before harvest at the close of the spring and fall trials, we sampled insect populations using a vacuum within each plot, six sections of row were vacuumed for two seconds blasts each. These samples were bagged and later analyzed under magnification to determine the number of individual pest species. Yellow sticky trap data can be found in Table 2 while data from vacuum samples can be found in Table 3. 

*Leaf damage data*: to determine the impact of treatments on flea beetle damage to arugula and mizuna greens, we measured the number of shot holes per unit leaf area. Prior to, but on the same day as harvest, we collected 10 arugula and 10 mizuna leaves sampled randomly from the middle row per plot. We measured leaf area using the application, LeafByte, on an Apple iPhone SE. The application LeafByte estimates total leaf area from photographs of leaves. After the leaf area was estimated, we counted the number of flea beetle damage holes per leaf and calculated the number of damage holes per cm^2^ of leaf area. Leaf damage data can be found in Table 3.

*Harvest data*: in the spring trial, we harvested arugula and mizuna on 20 May. Within the middle, experimental row, we cut all foliage within the central four feet of both crop species. Harvest was completed by cutting the entirety of the sampled foliage to the ground level using scissors. This sample was weighed, and all measurements were written down for analysis. In the fall trial, we harvested on 20 September. Two samples, each of which were a two foot length of row, was harvested for each crop species. This harvesting method was maintained consistently to prevent any biases. Harvest data can be found in Table 3.

To measure the bolting of the greens, we counted the total number of bolting stems within a ten foot length of row for each crop. Both arugula and mizuna mustard are harvested when they are young and only have a rosette of leaves near the ground. As both plant species mature, they will produce a shoot with a flowering head above the rosette of leaves. At this time, the leaves of the plant become very bitter tasting due to a transformation of the sugars within the leaf. This bitterness is unpalatable, rendering the produce unmarketable. We defined bolting for this study as any shoot with a flower head that was above the rosette of leaves. Harvest data can be found in Table 3. 

*Temperature data*: to determine potential differences in temperature between treatments, we placed temperature sensors (SpecWare 9, Spectrum Technologies, Inc., Aurora, IL, USA) within the Agribon and ProtekNet row cover treatments as well as in the uncovered control treatment. Using these field sensors, we followed temperature in the control treatment (*n* = 4), the fine mesh ProtekNet treatment (*n* = 4), and the spun-bonded polyethylene Agribon treatment (*n* = 4). We made the assumption that all uncovered treatments would experience the same temperature. Temperature data was only collected for the final five days of the spring trial as there was a delay in the arrival of the temperature sensors from the manufacturer. In the fall trial, we collected temperature data across the entire experiment. These sensors were placed within these three treatments to determine temperature differences between the row covers and bare ground. The sensor was placed in the middle of the row at a height of 10 to 12 inches above the ground. We calculated maximum and average temperatures and compared them across treatments. Temperature data can be found in Table 4. 

*Data analysis*: to determine the impact of row covers, essential oils, and organic and conventional insecticides on flea beetle abundance, damage, and crop yield we performed general linear mixed models (GLMM). We conducted analyses for the spring and fall trials independently given that the essential oil treatments differed between trials. For each dependent variable (flea beetle abundance, shot-holes per cm^2^, number of bolting stems, yield, maximum temperature, minimum temperature, average temperature), we incorporated treatment as a fixed effect within models. In order to nest the randomized block design into the model structure, we incorporated block as a random effect within models. Following GLMM, we performed Tukey’s post hoc tests to determine pairwise comparisons of different treatment levels if the overall treatment effect was significant. We tested all models for normality using a Shapiro–Wilk test on model residuals. If model residuals were not normally distributed, we transformed independent variables with square-root or log transformations until residual distributions met the assumptions of normality. For flea beetle abundance in vacuum samples of spring arugula, transformations did not improve the assumptions of normality. In this case, we analyzed data with a non-parametric Kruskal–Wallis test. All analyses were conducted in the program R (3.3.3) using the packages ‘LME4′, ‘stats’, and ‘emmeans’ (R Foundation, Vienna, Austria). Test statistics can be found in Table 5. 

## 3. Results

### 3.1. Spring Trial 

*Yellow sticky trap sampling of flea beetles:* in the spring trial, we observed 698 flea beetles on the yellow sticky traps. There was a significant effect of treatment on flea beetle abundance in the spring trial (Table 2 and Table 5). The Agribon treatment had fewer flea beetles than the control (*p* = 0.0001) and organic insecticide (*p* = 0.001) treatments. The ProtekNet treatment had fewer flea beetles than the control (*p* < 0.0001) and organic insecticide (*p* = 0.0009) treatments. The ProtekNet + rosemary oil treatment had fewer flea beetles than control (*p* < 0.0001) and organic insecticide (*p* = 0.001) treatments. The conventional insecticide treatment had fewer flea beetles than the control (*p* = 0.0005) and organic insecticide (*p* = 0.008) treatments. There were no other significant pairwise comparisons. 

*ProtekNet + Thyme oil treatment:* heavy phytotoxic burns were found 30 April on the ProtekNet + Thyme oil treatment. This was the day after the first essential oils spray was made. 25% of arugula plants were dead and an additional 70% of plants had lost most of their leaves; 73.5% of Mizuna died as a result of this spray with an additional 26.5% plants losing the majority of their leaves. Due to the magnitude of these burns, this treatment was removed from the project (Table 2 and Table 3).

#### 3.1.1. Arugula

*Vacuum sampling of flea beetles:* we collected 169 flea beetles from vacuum samples at the end of the growing season with 70% of these being *Phyllotreta striolata* and 9% being *P. bipustulata*. The remaining 21% were not identified. There were no significant pairwise comparisons within the vacuum samples (Table 3 and Table 5).

*Flea beetle damage:* there was a significant effect of treatment on flea beetle damage to arugula in the spring trial (Table 3 and Table 5). A Tukey post hoc test revealed that the control treatment had more damage than the conventional insecticide (*p* = 0.001), ProtekNet (*p* < 0.0001), Agribon (*p* < 0.0001), and ProtekNet + rosemary oil (*p* < 0.0001) treatments. The organic insecticide treatment had more damage than the conventional insecticide (*p* = 0.018), ProtekNet (*p* = 0.0001), Agribon (*p* < 0.0001), and ProtekNet + rosemary oil (*p* = 0.0001) treatments. Additionally, the conventional insecticide treatment had more damage than the Agribon (*p* = 0.044) treatment. There were no other significant pairwise comparisons. 

*Yield:* there was no significant effect of treatment on arugula yield in the spring trial (Table 3 and Table 5).

*Bolting:* there was a significant effect of treatment on arugula bolting in the spring trial (Table 3 and Table 5). A Tukey post hoc test revealed that the Agribon treatment had higher rates of bolting than the control (*p* = 0.0498), organic insecticide (*p* = 0.0254), and conventional insecticide (*p* = 0.0254) treatments. There were no other significant pairwise comparisons.

#### 3.1.2. Mizuna Mustard

*Vacuum sampling of flea beetles:* we collected 169 flea beetles from vacuum samples at the end of the growing season with 70% of these being *Phyllotreta striolata* and 9% being *P. bipustulata*. The remaining 21% were not identified. There was a significant effect of treatment on the abundance of flea beetles on mizuna mustard within the spring trial (Table 3 and Table 5). A Tukey post hoc test revealed that the ProtekNet + rosemary oil treatment had fewer flea beetles than the organic insecticide (*p* = 0.014) and conventional insecticide (*p* = 0.004) treatments. Additionally, the ProtekNet + rosemary oil treatment had marginally fewer flea beetles than the control (*p* = 0.071) treatment. There were no other significant pairwise comparisons. 

*Flea beetle damage:* there was a significant effect of treatment on the abundance of flea beetles on mizuna mustard within the spring trial (Table 3 and Table 5). A Tukey post hoc test revealed that the control treatment had more damage than the conventional insecticide (*p* = 0.0003), ProtekNet (*p* < 0.0001), Agribon (*p* < 0.0001), and ProtekNet + rosemary oil (*p* = 0.0001) treatments. The organic insecticide treatment had more damage than the conventional insecticide (*p* = 0.003), ProtekNet (*p* ≤ 0.0001), Agribon (*p* = 0.0003), and ProtekNet + rosemary oil (*p* = 0.001) treatments. There were no other significant pairwise comparisons. 

*Yield:* there was a significant effect of treatment on mizuna mustard yield in the spring trial (Table 3 and Table 5). A Tukey post hoc test revealed that the ProtekNet treatment had higher yields than the control (*p* = 0.0005) and organic insecticide (*p* = 0.001) treatments. There were no other significant pairwise comparisons.

*Bolting:* there was no bolting of mizuna mustard within the spring trial (Table 3 and Table 5).

### 3.2. Fall Trial 

*Yellow sticky trap sampling of flea beetles:* in the fall trial, we observed 3578 flea beetles on yellow sticky traps. There was a significant effect of treatment on the abundance of flea beetles within the fall trial (Table 2 and Table 5). The Agribon treatment had fewer flea beetles than the control (*p* = 0.0002), organic insecticide (*p* = 0.0001), neem (*p* = 0.0001), rosemary (*p* = 0.025), and conventional insecticide (*p* = 0.018) treatments. The ProtekNet treatment had fewer flea beetles than the control (*p* = 0.0035), organic insecticide (*p* = 0.0007), and neem (*p* = 0.0012) treatments. There were no other significant pairwise comparisons. 

*Temperature data:* there was a significant effect of treatment on temperature in the fall trial (Table 4 and Table 5). The Agribon treatment had a higher average temperature over the course of the trial than the control treatment (*p* = 0.008). The control treatment also had a lower max temperature than the Agribon (*p* = 0.049) and ProtekNet (*p* = 0.037) treatments. There were no other significant pairwise comparisons. 

#### 3.2.1. Arugula

*Vacuum sampling of flea beetles:* the vacuum sampling caught 726 flea beetles with 73% of these *P. striolata*, 6% *P. bipustulata*, 10% *P. cruciferae*, 8% Chaetocnema concinna, and the remaining 3% were not identified. The eggplant flea beetle, *Epitrix fuscula*, the tobacco flea beetle, *Epitrix fasciata*, and the pigweed flea beetle, *Disonycha glabrata*, were all found at low levels. There was a significant effect of treatment on the abundance of flea beetles within the vacuum samples (Table 3 and Table 5). The control treatment had significantly higher levels of flea beetles than the conventional (*p* = 0.001), ProtekNet (*p* = 0.0006), and Agribon (*p* = 0.0004) treatments. The neem oil treatment had more flea beetles than the conventional (*p* = 0.0087), ProtekNet (*p* = 0.0005) and Agribon (*p* = 0.0004) treatments. The organic insecticide treatment had more flea beetles than the ProtekNet (*p* = 0.05) and Agribon (*p* = 0.04) treatments. The organic insecticide treatment had marginally more flea beetles than the conventional treatment (*p* = 0.07). There were no other significant pairwise comparisons.

*Flea beetle damage:* there was a significant effect of treatment on flea beetle damage to arugula in the fall trial (Table 3 and Table 5). A post hoc test revealed that the control treatment had more damage than the conventional insecticide (*p* = 0.025), Agribon (*p* < 0.0001), and ProtekNet (*p* < 0.0001) treatments. The Agribon treatment had less damage than the organic insecticide (*p* < 0.0001), conventional insecticide (*p* = 0.001), rosemary oil (*p* = 0.0003), and neem oil (*p* < 0.0001) treatments. The ProtekNet treatment had less damage than the organic insecticide (*p* < 0.0001), conventional insecticide (*p* = 0.0005), rosemary oil (*p* = 0.0001), and neem oil (*p* < 0.0001) treatments. There were no other significant pairwise comparisons. 

*Yield:* there was a significant effect of treatment on arugula yield in the fall trial (Table 3 and Table 5). A Tukey post hoc test revealed that the rosemary oil treatment had lower yield than the organic insecticide (*p* = 0.0382), Agribon (*p* = 0.005), and ProtekNet (*p* = 0.0025) treatments. There were no other significant pairwise comparisons. Yield within the rosemary oil and neem oil treatments are believed to be impacted by light phytotoxicity burns first observed on 7 September. These burns were seen through the death of apical leaf tissue.

*Bolting:* There was no bolting of arugula within the fall trial.

#### 3.2.2. Mizuna Mustard

*Vacuum sampling of flea beetles:* the vacuum sampling caught 726 flea beetles with 73% of these *P. striolata*, 6% *P. bipustulata*, 10% *P. cruciferae*, 8% Chaetocnema concinna, and the remaining 3% were not identified. There was a significant effect of treatment on the abundance of flea beetles within vacuum samples (Table 3 and Table 5). The control treatment had significantly more flea beetles than the organic (*p* = 0.007), conventional (*p* < 0.0001), ProtekNet (*p* < 0.0001), Agribon (*p* < 0.0001), and rosemary oil (*p* < 0.0001) treatments. The neem oil treatment had more flea beetles than the organic insecticide (*p* = 0.015), conventional insecticide (*p* < 0.0001), ProtekNet (*p* < 0.0001), Agribon (*p* < 0.0001), and rosemary oil (*p* < 0.0001) treatments. The organic insecticide treatment had more flea beetles than the conventional insecticide (*p* < 0.0001), ProtekNet (*p* < 0.0001) and Agribon (*p* < 0.0001) treatments. The rosemary oil treatment had more flea beetles than the conventional insecticide (*p* = 0.007), ProtekNet (*p* = 0.007) and the Agribon (*p* = 0.007) treatments. There were no other significant pairwise comparisons.

*Flea beetle damage:* there was a significant effect of treatment on flea beetle damage to mizuna mustard in the fall trial (Table 3 and Table 5). A Tukey post hoc test revealed that the control treatment had more damage than the rosemary oil (*p* = 0.0058), organic insecticide (*p* < 0.0001), conventional insecticide (*p* < 0.0001), Agribon (*p* < 0.0001), and ProtekNet (*p* < 0.0001) treatments. The neem oil treatment was the same as the rosemary oil (*p* = 0.90) and the organic insecticide (*p* = 0.15) treatments. The conventional insecticide treatment had less damage than the neem oil (*p* < 0.0001), rosemary oil (*p* < 0.0001), and organic insecticide (*p* < 0.0001) treatments. The Agribon treatment had less damage than neem oil (*p* < 0.0001), rosemary oil (*p* < 0.0001), organic insecticide (*p* < 0.0001), and conventional insecticide (*p* < 0.0001) treatments. The ProtekNet treatment had less damage than neem oil (*p* < 0.0001), rosemary oil (*p* < 0.0001), organic insecticide (*p* < 0.0001), and conventional insecticide (*p* < 0.0001) treatments. There were no other significant pairwise comparisons. 

*Yield:* there was a significant effect of treatment on mizuna mustard yield in the fall trial (Table 3 and Table 5). A post hoc test revealed that the ProtekNet treatment had higher yields than the control (*p* = 0.0169), neem oil (*p* = 0.0034), and rosemary oil (*p* = 0.0038) treatments. There were no other significant pairwise comparisons. Yield within the rosemary oil and neem oil treatments are believed to be impacted by light phytotoxicity burns first observed on 7 September. These burns were seen through the death of apical leaf tissue.

*Bolting:* there was no bolting of mizuna mustard within the fall trial. 

## 4. Discussion

This study found that row covers are an effective method for controlling flea beetles within brassicaceous leafy greens. Both the fine-mesh row cover ProtekNet and the spun-bonded row cover Agribon had similar or gave better control of flea beetles than all other treatments. Both row cover treatments, without essential oil sprays, always reduced flea beetle damage significantly below the control and organic insecticide treatments (Table 3). One trend that was observed over the course of the two trials was that we had higher numbers of flea beetles within the fall trial and fewer beetles in the spring trial. This was likely due to cold snaps in the previous winter that killed off portions of the population. This information could be useful for growers when choosing when to grow brassicaceous greens and what management strategy to use for that season. In the fall trial, when flea beetle pressure was high, row cover treatments provided stronger flea beetle suppression than the conventional insecticide treatment. Furthermore, in the fall trial, ProtekNet was the only treatment to have significantly higher yields relative to the control treatment. To our knowledge, the only other study to compare the effectiveness of row covers with insecticide treatments corroborate our findings, with a few exceptions [14]. Andersen et al., [14] found that the row covers CovertanPro30 and Agril 17 gave the best control of flea beetles and the corresponding damage. They also found that carbaryl and spinosad lowered flea beetle numbers below that of the control. In their experiment, treatment with Kaolin and pyrethrin showed no difference from the control. Interestingly within their experiment, the conventional insecticide thiamethoxam had higher levels of flea beetles and damage than the control. Similarly, within our study, row covers served as the best control for flea beetles. Additionally, the conventional insecticide rotation of pyrethroids and neonicotinoids behaved similar to carbaryl in Andersen et al. [14]. 

Our two insecticide rotations behaved very differently in the field. The rotation of group 3A and 4A conventional insecticides provided intermediate control of flea beetles within both the spring and fall trials. Flea beetle damage within the conventional insecticide plots was statistically similar to the row cover treatments for both trials and crops. However, damage to leaves was higher in conventional insecticide plots and leaves may not be marketable due to the higher damage. High levels of flea beetles were found through vacuuming the conventional insecticide plots at the conclusion of the spring trial. One possible reason for the high numbers of flea beetles, which we observed in vacuum samples, is that there was a time delay between the last insecticide spray and the vacuuming (6 days). 

As opposed to conventional insecticides, organic insecticides provided very poor management of flea beetles. Over the course of the two trials and within both crops, the number of flea beetles and the corresponding crop damage rarely differed between organic insecticide treatments and the untreated control. Andersen et al. [14] found that their control treatment averaged 120 and 137 damage holes per leaf in the two Komatsuna (*Brassica rapa* var. perviridis) trials planted mid-June. If we convert our data to damage holes per leaf instead of holes per cm^2^, our study had lower damage levels than Andersen et al. [14]. Within the arugula control treatment, we found an average of 29 holes per leaf in the spring and 98 holes per leaf in the fall. Within the mizuna mustard control, we found an average of 28 holes per leaf in the spring and 79 holes per leaf in the fall. If we analyze just the arugula organic treatment, we found 21 holes per leaf in the spring and 57 holes per leaf in the fall. The high levels of damage to the leafy greens would lead to unmarketability. Additionally, these organic insecticides often must be purchased in impractical quantities for small farmers such as those in our region, who primarily sell their produce at local farmer’s markets. For these reasons, organic farmers need new management strategies that can be used within organic certification requirements.

Although plant essential oils have shown promising effects on pests in laboratory experiments, our study failed to observe benefits in the field. The results of the essential oil treatments within these two field trials were not promising due to phytotoxicity responses from the crop. Within the spring field season, the thyme oil passed through the row cover and completely killed both crops within a day of the first spray. This led to the removal of the thyme oil treatment for the remaining duration of the study. Essential oils were sprayed directly on the crop in the fall due to concerns of the cost effectiveness of spraying essential oils over row covers. In the fall field season, both the neem and the rosemary oils caused moderate drops in yield (Table 3) due to stunting associated with early season phytotoxicity. We believe that we saw phytotoxicity effects among the neem and rosemary oils due to spikes in temperature which heated the oil and scalded leaves. While researchers have evaluated thyme for insect repellency, it has also been studied, among other essential oils, for functionality as a herbicide [45]. Rosemary oil sprayed on ProtekNet within the spring field season did not have stunting due to phytotoxicity but did not provide better protection than the ProtekNet alone. We used a five percent concentration of essential oils and sprayed twice a week. This concentration and frequency were estimated as the maximum practical level of control that producers would use. When sprayed at lower concentrations or less frequently, these essential oils may behave differently. Further research is needed to determine the proper concentration and frequency to balance insect repellency and the negative effects to plant leaves.

We observed small numbers of flea beetles under the row cover and their corresponding damage. We suspect that most flea beetles entered the plots through small gaps where the row covers met the ground rather than entering through the material. The fine-mesh row covers did not have any rips over the course of the two trials and we do not believe that flea beetles entered through the mesh. However, we did find several small rips in the Agribon material. These rips were mended when found but flea beetles could have taken advantage of these holes before mending. To improve efficacy, care should be taken to minimize damage to the row cover, and the row cover should be held firmly to the ground with a weight such as a polyvinyl chloride pipe filled with water. Further studies that span several years are needed to understand the life expectancy of ProtekNet. Within our trial, the ProtekNet treatment had a positive effect on crop yield, particularly in the fall trial. This boost in yield could be due to a lack of stress from herbivores or abiotic conditions. Bolting was found within the spring trial arugula and was statistically higher in the Agribon treatment. We believe the higher levels of bolting were due to increased temperatures underneath the row cover. Increased bolting may occur when using row covers, particularly during warmer times of the year and in warmer climates than that of central Kentucky. 

Despite the strong effects of row covers on flea beetle damage in our study, there are a number of considerations that growers may want to make before adopting this practice. First, not all Brassicaceous crops may perform as well as arugula and mizuna under row covers. Furthermore, while we were able to compare the efficacy in the spring and fall in Kentucky, we were not able to compare multiple years or across regions. The efficacy of row covers may vary across time and across growing regions. Andersen et al. [14] found similarly positive results for Komatsuna grown under row covers in a multi-year study in Massachusetts, but both our study and Andersen et al. [14] could consider the challenges of scaling plot size up to commercial scale. For instance, all plots in our study were grown on bare ground and weeding was done manually while the row cover remained on. Future research must include a viable weed management system for commercial growers to adopt row covers on a large scale. Growers should also consider the cost of implementation. Many growers are already using Agribon within the field, and while ProtekNet is more expensive to cover the same area, ProtekNet should last many more seasons than Agribon. Future studies should include an analysis of longevity, cost analysis, and profitability of different row cover systems. 

While row covers were first introduced for season extension, mounting research is showing their effectiveness for insect exclusion. Row covers are most popular with organic and conventional growers in cropping systems that have a very low tolerance of insect damage. There is very low tolerance for insect damage where insects are causing direct damage to the produce such as in brassicaceous leafy greens [14], lettuce [32], and apples [46]. Additionally, there is very low tolerance for insect pests in crops where insects are vectors of plant pathogens such as whiteflies in tomatoes [47], cucumber beetles in cucurbits [38], and aphids (*Aphis gossypi* Glover, *Myzus persicae* Sulzer), whiteflies, and thrips species in hot peppers [48]. There are many types of row cover for various purposes. The ProtekNet row cover used within our experiment is a fine mesh, high-density, polyethylene netting manufactured for insect exclusion. The Agribon row cover we used is most often used as a frost blanket which allows producers to plant earlier in the spring and later in the fall. Our study shows that producers can use row covers, especially Agribon, for multiple reasons to maximize cost effectiveness. Row covers offer producers in the southeastern United States an effective method of flea beetle control that reduces their dependence on insecticides for conventional and organic production. 

## 5. Conclusions

In conclusion, we found that row covers performed better than organic insecticide treatments and often conventional insecticide treatments. Organic insecticides rarely controlled flea beetles better than the control. While previous studies have shown that essential oils have potential to repel pest insects, we found high levels of phytotoxicity in plots sprayed with an essential oil mixture. This phytotoxicity lowered yields and the quality of the greens. Conventional insecticide plots often had lower numbers of flea beetles than the control, but levels were high enough to cause damage that would seldom be acceptable to consumers. Variability in populations of flea beetles over the course of the season should affect producer’s decisions on when to grow brassicaceous greens and the pest management decisions they make. We found that row covers provide optimal control of flea beetles for the brassicaceous greens arugula and mizuna mustard.

## Figures and Tables

**Table 1 insects-11-00714-t001:** Spring and fall treatment descriptions.

**Spring Trial 2019**	
Control	No spray, no row cover
Organic insecticide	Rotation of Spinosad and Pyrethrin sprayed once per week ^1^
Conventional insecticide	Rotation of Pyrethroid and Dinotefuran sprayed once per week ^2^
Agribon row cover	Spun-bonded polyethylene row cover ^3^
ProtekNet row cover	25-gram fine mesh row cover ^4^
ProtkeNet and rosemary oil	ProtekNet row cover sprayed with rosemary essential oil twice a week ^5^
ProtkeNet and thyme oil	ProtekNet row cover sprayed with Thyme essential oil twice a week ^5^
**Fall Trial 2019**	
Control	No spray, no row cover
Organic insecticide	Rotation of Spinosad and Pyrethrin sprayed once per week ^1^
Conventional insecticide	Rotation of Pyrethroid and Dinotefuran sprayed once per week ^2^
Agribon row cover	Spun-bonded polyethylene row cover ^3^
ProtekNet row cover	25-gram fine mesh row cover ^4^
Rosemary oil	Rosemary essential oil applied directly onto greens (no row cover) twice a week ^6^
Neem oil	Neem essential oil applied directly onto greens (no row cover) twice a week ^6^

^1^ Pyganic Crop Protection 5.0_II_ (Pyrethrin, Valent U.S.A. Corporation, MGK, Minneapolis, MN, USA) and Entrust SC (Spinosad, Corteva Agriscience [Dow AgroScience], Indianapolis, IN, USA). ^2^ Mustang Maxx (Pyrethroid, Zeta-cypermethrin, FMC Corporation, Philadelphia, PA, USA) and Scorpion 35SL (Dinotefuran, Gowan Company, Yuma, AZ, USA). ^3^ (Agribon grade-20, Berry Plastics, Evansville, IN, USA). ^4^ (ProtekNet 25 gram, Dubois, Montreal, QC, USA). ^5^ treated twice a week with rosemary essential oil or thyme essential oil (Aura Cacia, Frontier Natural Products Co-op, Norway, IA) mixed at a 5% solution with 2.5% adjuvant (Nu Film P, Miller Chemical and Fertilizer, Hanover, PA, USA) and 92.5% water. ^6^ treated twice a week with rosemary essential oil (applied at same rate and mix as in 5) or neem oil was treated directly onto greens twice a week with neem oil prepared from a concentrate (Safer Brand, Woodstream Corporation, Lititz, PA, USA) 1 fluid ounce per gallon of water with 2.5% spreader sticker adjuvant (Nu Film P).

**Table 2 insects-11-00714-t002:** Effects of treatments on the number of flea beetles caught by yellow sticky traps (mean and standard error).

Spring 2019	
Treatment	No. Flea Beetles (Sticky Traps)
Control	16.63 (1.24) A
Organic insecticide	13.63 (1.70) A
Conventional insecticide	5.63 (0.99) B
Agribon row cover	3.13 (0.23) B
ProtekNet row cover	3.13 (0.55) B
ProtekNet + Rosemary oil	1.5 (0.31) B
ProtekNet + Thyme oil *	N/A (not applicable)
**Fall 2019**	
Control	47.81 (11.95) A
Organic insecticide	61.5 (15.38) A
Conventional insecticide	26.5 (6.63) AB
Agribon row cover	8.56 (2.14) C
ProtekNet row cover	3.88 (0.97) BC
Rosemary oil	24.63 (6.16) AB
Neem oil	52.81 (14.11) A

Common letters denote means are not significantly different from one another, as determined by Tukey’s honestly significant difference (HSD). Numbers in parenthesis are standard errors. * ProtekNet + thyme oil treatment removed due to death of plants after first essential oil spray.

**Table 3 insects-11-00714-t003:** Effects of treatment on number of flea beetles caught by vacuum (mean and standard error), flea beetle damage per unit leaf area (mean and standard error), yield in pounds per acre (mean and standard error), and the number of stems bolting (mean and standard error).

**Spring 2019**					
**Species**	**Treatment**	**Number Flea Beetles (Vacuum)**	**Damage (holes/cm^2^)**	**Yield (Pounds/acre)**	**Bolting (Stems)**
Arugula	Control	3.5 (1.89) A	0.87 (0.09) C	12,279 (1261) A	2.50 (1.55) A
	Organic insect.	3.5 (0.65) A	0.66 (0.07) C	14,006 (1447) A	2.25 (1.31) A
	Conventional insect.	6 (3.08) A	0.25 (0.04) B	12,046 (794 A	1.75 (0.85) A
	Agribon row cover	1.25 (0.95) A	0.07 (0.03) A	15,874 (1634) A	7.50 (2.36) B
	ProtekNet row cover	1.25 (0.75) A	0.07 (0.02) AB	16,948 (1027) A	3.75 (2.10) AB
	ProtekNet + Rosemary	0.75 (0.25) A	0.08 (0.02) AB	13,540 (1447) A	3.25 (1.11) A
	ProtekNet + Thyme *	N/A	N/A	N/A	N/A
Mizuna	Control	5.25 (2.25) ab	1.50 (0.12) b	10,178 (2288) a	0
	Organic ins.	7.5 (3.12) b	1.09 (0.11) b	10,878 (2334) a	0
	Conventional ins.	7.5 (2.53) b	0.25 (0.05) a	14,847 (2941) ab	0
	Agribon row cover	2.25 (0.75) ab	0.13 (0.02) a	17,415 (1074) ab	0
	ProtekNet row cover	2.25 (1.03) ab	0.09 (0.02) a	21,617 (1587) b	0
	ProtekNet + Rosemary	1.25 (0.95) a	0.17 (0.03) a	16,247 (2101) ab	0
	ProtekNet + Thyme *	N/A	N/A	N/A	N/A
**Fall 2019**					
**Species**	**Treatment**	**Number Flea Beetles (Vacuum)**	**Damage (holes/cm^2^)**	**Yield (Pounds/acre)**	**Bolting (Stems)**
Arugula	Control	57.25 (12.85) B	3.22 (0.24) C	9011 (560) AB	0
	Organic ins.	37.25 (11.20) B	2.01 (0.21) BC	9571 (840) A	0
	Conventional ins.	5.50 (2.25) A	1.24 (0.15) B	8544 (794) AB	0
	Agribon row cover	1.50 (0.29) A	0.28 (0.07) A	11,672 (1213) A	0
	ProtekNet row cover	3.25 (0.85) A	0.17 (0.03) A	12,372 (1121) A	0
	Rosemary oil	29.50 (2.84) AB	1.41 (0.15) BC	3968 (327) B	0
	Neem oil	47.25 (8.20) B	2.19 (0.21) BC	7143 (1307) AB	0
Mizuna	Control	77.00 (12.25) c	5.12 (0.37) d	7844 (420) b	0
	Organic ins.	50.25 (6.34) b	2.86 (0.36) c	9104 (607) ab	0
	Conventional ins.	7.75 (1.70) a	1.32 (0.18) b	9711 (420) ab	0
	Agribon row cover	6.75 (1.31) a	0.27 (0.04) a	11,905 (514) ab	0
	ProtekNet row cover	6.25 (0.63) a	0.19 (0.02) a	15,407 (2101) a	0
	Rosemary oil	32.75 (3.42) b	2.99 (0.27) c	6443 (1214) b	0
	Neem oil	74.25 (0.16) c	3.58 (0.33) cd	6303 (980) b	0

Common letters denote means are not significantly different from one another, as determined by Tukey’s HSD Capitalization used for arugula and lowercase letters used for mizuna mustard. Numbers in parenthesis are standard errors. * ProtekNet + thyme oil treatment in spring trial removed due to death of plants.

**Table 4 insects-11-00714-t004:** Effect of treatment on temperature (Fahrenheit) in the fall trial.

Fall Temperature 2019		
Treatment	Maximum	Minimum
Agribon	109.6 (0.97) B	76.3 (0.14) B
ProtekNet	110.1 (0.78) B	75.8 (0.21) AB
Control	103.6 (0.82) A	75.3 (0.30) A

Common letters denote means are not statistically significant. Numbers in parenthesis are standard errors.

**Table 5 insects-11-00714-t005:** Statistical analysis of effect of treatment on number of flea beetles found, flea beetles damage per unit leaf area, crop yield, and the number of stems bolting.

**Spring 2019**			
**Species**	**Treatment Effect**	**F**	***p***
Both	No. flea beetles (sticky traps)	22.2	<0.001
Arugula	No. flea beetles (vacuum)	9.9 (H statistic *)	0.08
Arugula	Damage (holes/cm^2^)	27.8	<0.001
Arugula	Yield (lbs)	2.6	0.07
Arugula	Bolting	3.8	0.02
Mizuna	No. flea beetles (vacuum)	4.2	0.01
Mizuna	Damage (holes/cm^2^)	21.9	<0.001
Mizuna	Yield (lbs)	4.8	0.008
Mizuna	Bolting	-	-
**Fall 2019**			
**Species**	**Treatment Effect**	**F**	***p***
Both	No. flea beetles (sticky traps)	2.1	0.15
Both	Temperature maximum	6.8	0.02
Both	Temperature minimum	11.1	0.01
Arugula	No. flea beetles (vacuum)	9.7	<0.001
Arugula	Damage (holes/cm^2^)	26.1	<0.001
Arugula	Yield (lbs)	5.2	0.002
Arugula	Bolting	-	-
Mizuna	No. flea beetles (vacuum)	31.8	<0.001
Mizuna	Damage (holes/cm^2^)	124.9	<0.001
Mizuna	Yield (lbs)	5.4	0.002
Mizuna	Bolting	-	-

* Kruskal–Wallis test was used for this statistic.

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
