# Peer review of "The Impact of Plant Essential Oils and Fine Mesh Row Covers on Flea Beetle (Chrysomelidae) Management in Brassicaceous Greens Production"

_insects, 2020, doi:10.3390/insects11100714_

Round 1

Reviewer 1 Report

I think this is a well-written and interesting applied paper.  It is a very pertinent topic in Europe as well as North America.  It will be of interest to those concerned specifically with these pests.  There has been some work on flea beetle management in the UK (reported on the AHDB web site) and flea beetle damage in relation to use of crop covers on radish to manage cabbage maggot has been reported in a paper in Crop Protection.  

I have made a number of suggestions for small changes to increase the 'precision' of the descriptions in the paper. (file attached)  I think possibly the Discussion could have been split into more paragraphs.  

One thought is about the use of the term 'cruciferous' as I think that Cruciferae has been replaced by Brassicaceae?  In which case I think it would be 'brassicaceous'?

My main concern is that I believe that some of the Discussion section should have been covered in the Results - especially with regard to phytotoxicity.  So I think this should be re-drafted.  I checked again to see if I had missed it but i don't think so!

Reviewer 2 Report

The manuscript is well written and supplies data about the efficacy of essential oils and row covers compared to organic and conventional pesticide sprays for control of flea beetles attacking cruciferous leafy greens in Kentucky; the information is new and therefore the publication of the manuscript is encouraged after a minor revision taking into consideration some of the shortcomings I have pointed out below.

My main concern is that the authors are extrapolating the applicability of their results beyond what the design supports. It is important that the authors state what is obvious with regard to generalizing their results to the larger question of row cover treatment benefits in cruciferous crops. The applicability could differ depending on a crop type, which the authors do indeed discuss.  There are others though; these are only one year of data from two seasons and small plots in a single field so the inference power of the paper is very limited, but authors do not acknowledge this detail at all and need to be more forthcoming. What about other weather, planting dates, rotation schemes, regions, etc.?  How confident can I be (as a farmer) that this result will hold in MY fields THIS year? What are the economic risks to ME if these results were mostly location and/or year specific? Yes, the treatments were replicated among small adjacent plots; but it is basically a case of pseudo-replication where the experiment has not been replicated in different locations, or at least different fields within a general location. We don’t have any idea how variable/repeatable the results will be in other places. This is a critical limitation of the study, and the authors must concede and discuss this. The consequences of getting this wrong will affect real people and livelihoods. So I am suggesting to the authors to tone-down the language a little and admit that there are still substantive uncertainties to be considered, including uncertainty as to how generalizable the results are to other locations, regions, and years.  Instead, the importance of the results is that they emphasize the need to do such studies in other areas and to understand better the benefits and variability in benefits derived from use of row covers in these crops. This is not to diminish the data gathered in this study, they are of value. But it is important for the authors not to overgeneralize, and to warn the reader (including regulatory agencies…) against doing so as well.
